# Side Effect/Complication Risk Related to Injection Branch Level of Chemoembolization in Treatment of Metastatic Liver Lesions from Colorectal Cancer

**DOI:** 10.3390/jcm10010121

**Published:** 2020-12-31

**Authors:** Marcin Szemitko, Elzbieta Golubinska-Szemitko, Ewa Wilk-Milczarek, Aleksander Falkowski

**Affiliations:** 1Department of Interventional Radiology, Pomeranian Medical University, Al. Pow. Wielkopolskich 72, 70-111 Szczecin, Poland; bakhis@hot.pl; 2Department of General and Dental Diagnostic Imaging, Pomeranian Medical University, Al. Pow., Wielkopolskich 72, 70-111 Szczecin, Poland; e.golubinska@gmail.com (E.G.-S.); ewawilkmilczarek@gmail.com (E.W.-M.)

**Keywords:** colorectal cancer, hepatic metastases, chemoembolization, DEB-TACE

## Abstract

Purpose: Transarterial chemoembolization with drug eluting beads (DEB-TACE) loaded with irinotecan despite having proven efficacy in the treatment of unresectable liver metastases in the course of colorectal cancer (CRC) does not have an established consistent method. In particular, there are discrepancies in the branch level at which microspheres are administered. Lobar embolization supplies microspheres to all vessels supplying a metastatic lesion but exposes the entire liver parenchyma to negative effects from microsphere irinotecan. Superselective chemoembolization compromises healthy liver parenchyma less but may omit small vessels supplying metastatic lesions. Objective: Assessment of the risk of complications and the severity of postembolization syndromes with CRC metastatic liver lesion chemoembolization with irinotecan-loaded microspheres, according to branch level of chemoembolization. Patients and methods: The analysis included 49 patients (27 female/22 male) with liver metastases in the course of CRC, who underwent 192 chemoembolization treatments (mean 3.62 per patient) with microspheres loaded with 100 mg irinotecan. The procedures were performed according to an adopted schema: alternating the right and left lobe of the liver at 3-week intervals. The severity of postembolization syndrome (PES) and the presence of complications were assessed according to the branch level of chemoembolization; microspheres were administered at the branch level of lobar, segmental, or subsegmental arteries. Assessment of adverse events was performed according to the standards of the Cancer Therapy Evaluation Program Common Terminology Criteria for Adverse Events, Version 5.0. Results: The median survival of all patients from the start of chemoembolization was 13 months. With 192 chemoembolization sessions, 14 (7.3%) serious complications were found. The study showed no significant relationship between the branch level of embolizate administration and the presence of complications (*p* = 0.2307). Postembolization syndrome was diagnosed after 102 chemoembolization treatments, i.e., 53.1% of treatments. A significant correlation was found between the severity of the postembolization syndrome and the branch level of embolization treatment (*p* = 0.00303). The mean PES severity increased from subsegmental through segmental to lobar administration. Conclusion: Chemoembolization using Irinotecan-loaded microspheres was relatively well tolerated by patients and gave a low risk of significant complications, which did not change with the branch level of microsphere administration. However, an association was found between the branch level of chemoembolization and the severity of postembolization syndrome. Further research is needed to determine the most effective DEB-TACE chemoembolization technique.

## 1. Introduction

Colorectal cancer (CRC) is the fourth most common cancer in the human population [1]. Around 50–60% of patients diagnosed with CRC develop colorectal hepatic metastases (CRHM). Surgical resection of the affected liver parenchyma, which is one possible therapeutic method, offers the longest survival time for patients [2]. Unfortunately, in about 80% of CRHM patients, the location and number of lesions make it impossible to resect them or there are other contraindications present for this type of surgery [3]. In such patients, in addition to standard systemic chemotherapy, it is possible to use chemoembolization, thermoablation, microwave ablation, and/or cryoablation. Chemoembolization with microspheres (DEB-TACE) loaded with irinotecan is a promising technique and there are reports of greater efficacy compared to systemic chemotherapy [4,5]. The advantage of chemoembolization is that it embolizes metastatic lesions through the delivery of high doses of chemotherapeutic agents directly to the lesions, while also minimizing systemic exposure. There are studies that have confirmed that chemoembolization with irinotecan-loaded microspheres is effective, even if previous systemic chemotherapy has failed [6].

However, there are some discrepancies in the current literature regarding the recommended branch level of microsphere administration such as the optimal microcatheter position within the branches of the hepatic artery. Most authors recommend widespread embolization of the liver parenchyma with the administration of microspheres at the branch level of the lobar artery [7]. However, there are reports that confirm the beneficial effects present with chemoembolization in both selective (lobar) or superselective (segmental and subsegmental) administration [8].

Superselective chemoembolization exposes less of the uninvolved liver parenchyma to the damaging effects of irinotecan-loaded microspheres but conversely may result in the omission of small vessels supplying metastatic lesions. Lobar embolization, on the other hand, allows for the delivery of microspheres to virtually all vessels supplying metastatic lesions, but exposes the entire liver parenchyma to the negative impact of irinotecan-loaded microspheres, which may result in more frequent occurrence of side effects and complications including PES (postembolization syndrome). These relationships have been assessed previously with chemoembolization in the course of hepatocellular carcinoma (HCC) performed with microspheres loaded with doxorubicin [9]. The aim of the present study is to assess the relationship between the administration site of irinotecan microspheres (the branch level of embolization) and the subsequent presence of complications and the severity of postembolization syndrome (PES) in the treatment of liver metastatic lesions from colorectal cancer.

## 2. Materials and Methods

This retrospective study included the chemoembolization of hepatic metastatic lesions in the course of CRC. The study was authorized by the Bioethics Committee of the Pomeranian Medical University. Procedures were performed in 49 consecutive patients (27 women and 22 men from November 2016 to December 2018). A total of 192 chemoembolization procedures were performed with microspheres loaded with 100 mg irinotecan. Qualification for the procedure was determined after consultation with an oncologist, based on computed tomography (CT) and/or magnetic resonance imaging (MRI) of the abdominal cavity and histopathological and laboratory results. Indications were established according to the recommendations of the European Society of Medical Oncology (ESMO) and the National Comprehensive Cancer Network (NCCN) in respect to the presence of an advanced disease that is unsuitable for surgery and/or is resistant to systemic therapy [1]. Inclusion criteria was the confirmed diagnosis of CRC with unresectable, dominant liver metastases, progression after prior systemic chemotherapy, Eastern Cooperative Oncology Group (ECOG) performance status < 2, age over 18 years old, and informed consent of the patient.

Exclusion criteria for the study were ECOG ≥ 2, hepatic failure (Child-Pugh B and C); ascites, bilirubin levels above 3 mg/dL, increase in ALT and AST >3× upper limit of normal, involvement of more than 50% of the liver parenchyma, renal failure (creatinine above 2 mg/dL), high risk of gastrointestinal bleeding from esophageal varices in the case of portal hypertension, thrombocytopenia below 50,000 mm^3^, or allergy to contrast.

The therapeutic scheme assumed an initial two procedures if there was involvement of only one lobe of the liver or four procedures in the case that both lobes were involved. Treatments were performed at 3-week intervals with alternating embolization of the branches of the right or left hepatic artery (the first designated at random) plus additional arteries supplying the liver lesions. Irinotecan solution (Accord Healthcare, Newcastle upon Tyne, UK) at a dose of 100 mg was mixed with a 2 mL syringe of 100 μm Embozene TANDEM (CeloNova Biosciences, now Boston Scientific, Marlborough, MA, USA) according to the manufacturer’s instructions. After the irinotecan loading was completed, the supernatant was ejected and the irinotecan-loaded microspheres were mixed with 10 mL of contrast material (iodixanolum 320 mg I/mL).

The procedure was performed by interventional radiologists with skills certificates and at least 6 years of experience in interventional radiology.

Each patient received prophylactic antibiotics, steroids, and proton pump inhibitors on both the day before and on the day of procedure. On the day of procedure, they additionally received an antiemetic and an infusion of 1000 mL of 0.9% NaCl.

### 2.1. Operation Procedure

Chemoembolization was initiated by puncture of the right or left common femoral artery. Then, the celiac trunk was catheterized (in the case of an atypical anatomical variant of the hepatic artery visible in the previous CT scan, the superior mesenteric artery was also catheterized) using a combination of SIM 5F catheter (Cordis/Johnson & Johnson, Miami, FL, USA), arteriography, and Dyna-CT examination (cone-beam CT). After arteriography, liver vascularization and metastatic changes were assessed. Subsequently, the lobar branches of the right or left hepatic artery and segmental or subsegmental branches were superselectively catheterized using a PROGREAT^®^ 2.7F microcatheter (Terumo, Tokyo, Japan), as follows.

In the case of a small and single focal lesion, supplied from one or two subsegmental branches, superselective chemoembolization was performed by placing the tip of the catheter in these arteries. Large metastatic lesions or numerous metastatic lesions supplied by multiple vessels extending from segmental arteries were embolized at the branch level of segmental arteries and in the case of lesions supplied directly from lobar branches at the level of lobar vessels. Prior to the administration of the embolizate, selective arteriography was performed to verify the position of the microcatheter and to exclude fistulas; the image was then archived in the “picture archiving and communications system” (PACS). Each administration of microspheres was preceded by microcatheter injection of 2–5 mL lidocaine. The mixture of microspheres and contrast agent was injected through the microcatheter by hand at a speed of around 1 mL/min, taking care to avoid reflux proximal to the catheter tip. Once near-stasis was achieved at the level of the vessel where the microcatheter tip was located, the administration of microspheres was stopped. Then the microcatheter was repositioned and subsequent lesions were embolized. During the procedure, the patient was given analgesic treatment under the care of an anesthesiologist. Intra- and postprocedural pain was usually controlled by continuous infusion of opioids (20 mg morphine/24 h) and nonsteroidal anti-inflammatory agents (i.e., paracetamol 2000 mg/24 h). Prophylactic antiemetics (ondansetron 8 mg i.v. and dexamethasone 8 mg i.v.) and antibiotic (cefazolin 1 g i.v.) were administered twice a day. After the surgery, the patient was admitted to the oncological surgery ward where, depending on their needs, analgesic and antiemetic treatment was continued.

In 152 DEB-TACE treatments, the patients were discharged from the hospital the day after the procedure and in another 32 treatments, 2 days after the procedure due to severe postembolization syndrome.

In 8 patients, hospitalization time was extended by 3–7 days due to complications or severe postembolization syndrome.

The extent of embolization was independently assessed on the basis of archived images by three interventional radiologists, each of whom has had at least 6 years of experience in oncological interventional radiology. Three branch levels of microcatheter tip position, at the initial point of microsphere administration, were defined:Superselective in a subsegmental branch of the hepatic artery;Superselective in a segmental branch of the hepatic artery;Selective in a lobar branch of the hepatic artery.

### 2.2. Assessment of Complications

The primary endpoint was the incidence and severity of PES. The presence of adverse events was a secondary endpoint. The assessment of complications and postembolization syndrome was based on the observation of the patient during hospitalization, in the interview 7 days after surgery, and at examination after 14–21 days.

Adverse reactions and complications occurring periprocedurally and within 30 days after the procedure were assessed according to the standards and terminology of the Cancer Therapy Evaluation Program Common Terminology Criteria for Adverse Events, Version 5.0.

Pain was assessed during and after the procedure on a scale of 0–10 points, i.e., 0 with no pain and 10 with very severe pain.

Postembolization syndrome was assessed according to an adopted scale (9:0) with 0 for no symptoms of postembolization syndrome; (1) moderately severe postembolization syndrome, not requiring additional treatment: moderate pain (1–5 points), increased body temperature up to 38 °C; and (2) severe postembolization syndrome requiring additional treatment: severe pain (6–10 points), fever over 38 °C, nausea and vomiting. These data were saved in a spreadsheet (Excel 2007; Microsoft, Washington, DC, USA).

### 2.3. Patient Characteristics

All patients enrolled in the study had unresectable CRC metastases in the liver, and five patients also had lung metastases with the liver being the predominant site of metastasis. In 39 patients, metastases were found in both liver lobes (mean 8.4 metastases per lobe), whereas in 10 patients, metastases were limited to one lobe (mean 2.1 metastases). In 44 patients, the degree of involvement of the liver parenchyma was below 25%, and only in five patients, it was within the range of 25–50% (with none above).

Each patient had been previously treated with at least one systemic chemotherapy regimen and 38 patients had received two or three lines of chemotherapy with evidence of progression (Table 1).

### 2.4. Feasibility of Chemoembolization

A total of 192 chemoembolization procedures were performed in 49 patients (27 women and 22 men), on average 3.62 per patient (Table 2). The technical success was 100%. In 10 patients, 20 chemoembolization treatments were performed with one lobe only. The remaining 39 patients underwent 172 chemoembolization procedures with two affected lobes.

### 2.5. Statistical Analyses

Descriptive statistics of the studied variables were given as an arithmetic mean and standard deviation or as a median and range. The relationships between branch levels of embolization and the presence or absence of complications or the severity of PES were assessed using the Pearson’s Chi-squared test. A *p*-value of < 0.05 was considered significant (two-tailed tests were used). Overall survival (OS) was calculated using the Kaplan–Meyer method from the date of the patient’s first DEB-TACE treatment to either the date of the last follow-up visit for that patient or the patient’s death. All statistical analyses were performed using a commercial program (Statistica, ver. 13.1; www.statsoft.pl; StatSoft Polska, Krakow, Poland).

## 3. Results

The median survival time after the chemoembolization treatments was 13 months. The 1-year survival of chemoembolization was 62% and the 2-year survival was 33%.

Postembolization syndrome was diagnosed after 102 DEB-TACE procedures, which constituted 53.1% of procedures. In 67 cases (65.7%), PES was mild, while 35 cases (34.3%) had high-grade PES requiring additional treatment. The highest intensity of PES occurred in lobar and segmental embolization (Table 3). A statistically significant correlation was found between the branch level of embolization and the severity of postembolization syndrome (chi-squared value = 15.989, degrees of freedom (df) = 4, *p*-value = 0.00303). The mean PES severity increased (Table 4) from subsegmental through segmental to lobar administration.

### Adverse Events

With 192 chemoembolizations, 14 (7.3%) serious complications were found. The most complications were found with the administration of embolizate at the level of the lobar arteries. There was no significant correlation between the branch level of embolization and the presence of complications (chi-squared = 2.9331, df = 2, *p*-value = 0.2307; Table 5). In two cases, an anaphylactic reaction occurred with moderate hypotension, skin redness, and coughing occurring during the procedure; this resolved after the intervention of the anesthetic team. Two other patients with severe and prolonged pain in the right upper quadrant showed signs of cholecystitis (without bilirubin increase) in ultrasound, which resolved after conservative treatment. In another two patients, features of liver decompensation with ascites were found. One patient experienced a septic episode with liver abscess 2 weeks after the last treatment, which was successfully treated by antibiotic therapy. Follow-up imaging studies showed signs of dilatation of the bile ducts in two patients as a result of damage, (Figure 1). Two patients had occlusion of the right or left branch of the hepatic artery (Figure 2). Three patients, 21 days after surgery, had leukopenia <2000 mm^3^, requiring the date of the next transcatheter arterial chemoembolization (TACE) session to be shifted by an additional week or two. There were no deaths in the periprocedural period or within 30 days of the procedure.

## 4. Discussion

Intra-arterial hepatic chemoembolization (TACE) is a locoregional therapy most commonly used in the treatment of inoperable hepatocellular carcinoma (HCC) [10] and neoplastic liver metastases and consists of injecting chemotherapy, mixed with embolization microspheres (DEB-TACE) or Lipiodol (cTACE), into the branches of arteries supplying neoplastic lesions. The expected advantage of TACE is that higher drug concentrations can be delivered to the tumor with reduced systemic exposure compared to systemic chemotherapy.

Studies using DEB-TACE chemoembolization show a risk of serious complications estimated at 1.6–11% [11] and a 30-day mortality of up to 1% [12]. The most common serious complications include liver failure and cholecystitis. Other complications that can occur are increased blood pressure, leukopenia, thrombocytopenia, thrombosis of a branch of the hepatic artery, and damage to the peribiliary plexus, accompanied by segmental dilatation of the bile ducts. Less common complications are cardiac episodes (myocardial infarction and arrhythmias), renal failure, intravascular complications, tumor rupture, liver abscesses, and complications related to the migration of embolization material. Migration out of liver blood vessels to the gastroduodenal artery is associated with the risk of pancreatitis or inflammation and bleeding from the gastric and duodenal mucosa; migration to the lungs can cause pulmonary embolism.

Numerous studies have shown that chemoembolization using irinotecan-loaded microspheres is relatively well-tolerated by patients. The most frequently observed side effects after chemoembolization includes pain in the right upper abdomen, fever, nausea, and/or vomiting [13]. Right upper abdominal pain is, according to most patients, the cause of the greatest discomfort after treatment due to its severity. These symptoms are included in the postembolization syndrome, which is the most common side effect of chemoembolization.

While with HCC chemoembolization, the “gold standard” is total embolization of well-vascularized tumors including the embolization of the afferent vessels [14], in the case of liver metastatic lesions, the main goal is to deliver a chemotherapeutic agent (in the present study: irinotecan) to the metastatic lesions. Irinotecan’s pharmacokinetics include conversion into the active metabolite SN-38 (7-ethyl-10-hydroxycamptothecin) in liver hepatocytes.

The current literature is dominated by an argument for lobar administration of irinotecan-loaded microspheres in order to deliver a chemotherapeutic agent to all liver lesions [15]. However, arguments for superselective administration of chemoembolizate are being raised more and more often, especially in the case of single metastatic lesions, with the argument that the healthy liver parenchyma would be less exposed to the toxic effects of irinotecan and that a higher dose can be delivered with a greater concentration of the active metabolite of irinotecan within the tumors [16,17]. In our study, we assessed several location variants of liver chemoembolization in terms of their side effects. The observed overall complication rate was low, which is consistent with conclusions from large multicenter studies regarding the safety of chemoembolization treatments [18]. We did not notice a statistically significant difference in the number of severe complications between lobar, segmental, or subsegmental administrations of embolizate. We have shown, however, that, in contrast to superselective embolization, lobar administration of irinotecan-loaded microspheres resulted in greater frequencies of severe postembolization syndrome.

Factors that predispose to a higher rate of complications, apart from systemic factors and the efficiency and functioning of the liver, include the dose of irinotecan administered, the size of the microspheres, and the achievement of long-term stasis in the liver vessels. In the present study, treatment factors were standardized. Moreover, the appropriate qualification of patients for the procedure and the procedural techniques are of great importance. We accepted qualifications of patients according to ESMO and NCCN rules and used current standards of chemoembolization with metastatic lesions, such as intra-arterial injection of lidocaine each time before administration of embolizate, its slow administration, and avoiding reflux and excessive stasis. The lack of a significant relationship between the branch level of embolizate administration and the occurrence of complications, as shown in our study, allows, depending on needs, the use of both techniques, e.g., in the case of single and large lesions, superselective embolization and in the case of multiple or multilevel vascularized lesions, lobar embolization. An issue that requires further research is the comparison of the effectiveness of the abovementioned methods of embolization.

The present study has shown that the severity of PES depends on the branch level of chemoembolization. This shows that the more extensive the type of embolization performed, the greater the possibility of PES occurrence. The possibility of using superselective embolization in selected cases will allow avoidance of PES and could reduce the discomfort of seriously ill patients.

## 5. Conclusions

Chemoembolization with irinotecan-loaded microspheres is relatively well tolerated by patients and has a low risk of significant complications, which is not affected by the branch level of microsphere administration. However, the branch level of chemoembolization affects the severity of PES. Further research is needed to determine the most effective DEB-TACE chemoembolization technique.

## 6. Limitations

(1) The small sample size. (2) The study was based on a retrospective analysis, not a randomized trial. (3) In addition, we used all-cause mortality, which made it impossible to distinguish cancer progression from other causes of death.

## Figures and Tables

**Figure 1 jcm-10-00121-f001:**
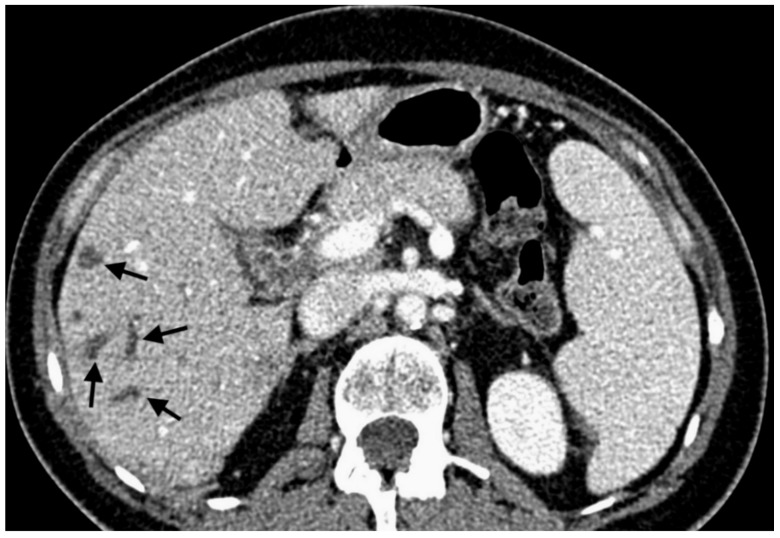
Contrast-enhanced multidetector computed tomography scan showing bile duct dilatation (black arrows) after transarterial chemoembolization with drug-eluting microspheres preloaded with irinotecan.

**Figure 2 jcm-10-00121-f002:**
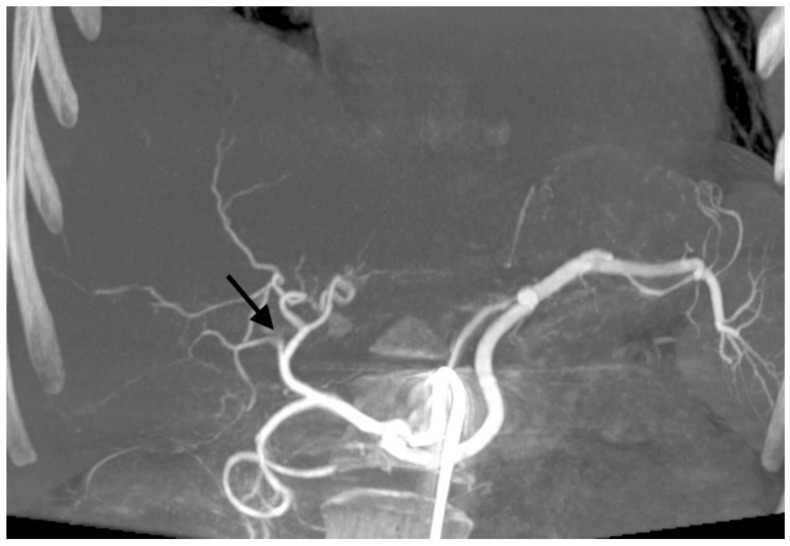
Cone-beam computed tomography showing occlusion of the right branch of hepatic artery (black arrow) after transarterial chemoembolization with drug-eluting microspheres preloaded with irinotecan.

**Table 1 jcm-10-00121-t001:** Patient characteristics.

Parameter	Value
Number of patients	49
Age, median (range)	68.4 (32–85)
Gender, male/female (*n*)	22/27
ECOG status: 0/1 (*n*)	36/13
Colon/rectal cancer (*n*)	37/12
Bilobar/unilobar metastases (*n*)	39/10
Number of liver metastases, median (range)	7.04 (1–19)
Extent of liver involvement (*n*, <25%/>25%)	44/5
Extrahepatic metastasis (*n*, %)	5 (10.2%)
**Number with line of prior systemic chemotherapy:** None 1 2 3	0 11 28 10
Prior liver surgery/ablation (*n*)	9/0
Prior locoregional therapy (*n*)	0

**Table 2 jcm-10-00121-t002:** Technical details of therapy with drug-eluting microspheres (100 μm) loaded with irinotecan.

Parameter	Value
Total number of treatments	192
Number of treatments per patient: mean (range)	3.62 (1–8)
**Number with each treatment selectivity:** Lobar Segmental Subsegmental	75 89 28
**Number with treatment location:** Right Left	99 93

**Table 3 jcm-10-00121-t003:** Branch level of catheter tip associated with incidence and severity of postembolization syndrome (PES).

Branch Level of Cathete Tip	Number with Mild Symptoms of PES	Number with Severe Symptoms of PES
Subsegmental artery	7	0
Segmental artery	30	13
Lobar artery	30	22
**Total (*n*, %)**	**67 (65.7%)**	**35 (34.3%)**

**Table 4 jcm-10-00121-t004:** Branch level of catheter tip with average severity of postembolization syndrome (PES; on a Scheme 0) compared using Pearson’s Chi-squared test (significance at *p* < 0.05).

Branch Level of Tip Catheter	Average Severity of PES
Subsegmental	1.25
Segmental	1.63
Lobar	1.89
**Total average**	**1.67 (*p*-value = 0.003034)**

**Table 5 jcm-10-00121-t005:** Branch level of catheter tip with the incidence of adverse events, compared using Pearson’s Chi-squared (significance at *p* < 0.05).

Branch Level of Catheter Tip	Number with Each Stage		
	G 2	G 3	Total
Subsegmental artery	2	1	3
Segmental artery	1	4	5
Lobar artery	6	0	6
**Total**	**9**	**5**	**14 (*p*-value = 0.2307)**

## Data Availability

The data presented in this study are available on request from the corresponding author.

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
