# Peer review of "Side Effect/Complication Risk Related to Injection Branch Level of Chemoembolization in Treatment of Metastatic Liver Lesions from Colorectal Cancer"

_jcm, 2020, doi:10.3390/jcm10010121_

Round 1

Reviewer 1 Report

xCertainly the technical aspect of which type of artery to cannulate is very important and all in all little known.There are several conceptual inaccuracies that need to be improved. First of all, it must be remembered that DEBIRI means DC-Beads while if you use Embozene or other microspheres this is a Deb-TACE. It seems useful to specify it even if very often DEBIRI is used to indicate a TACE that uses any type of microspheres. Anyway no one has ever shown that all the types of microspheres used have similar characteristics, indeed they always show different structural characteristics and loading capacity, compressibility and loading and unloading times of the drug. For this reason it is necessary to replace the word DEBIRI with Deb-TACE adopting Embozene or rather and better not using the trade name but use "Deb-TACE with microspheres of hydroGel core with polyzene-F coating ". Another point to be better defined is the infusion rate. In materials and metheod what does it mean "slowly"? Pls report how many ml of solution are infused for each minute and what dilution the vial of Embozene has. This is crucial because we know that PES correlates with these parameters probabibly much more than the selection of the diameter of the lobar, or segmental, vessels.There are unclear terms: line 109-110 eve? surgey? 137: most? pls specify the number, 291: lignocaine? 187 better anesthesist, 247 forget lipiodol,pls specify differences c-TACE Deb-TACE? references 15  tumordestruction pls detache . The observation time at 1-7 days is too long between 1 and 7 because we see the increase in transaminases after 2-3 days, pls add data from 1 to 5 days.

Author Response

Thank you for the critiques and suggestions.

1.There are several conceptual inaccuracies that need to be improved. First of all, it must be remembered that DEBIRI means DC-Beads while if you use Embozene or other microspheres this is a Deb-TACE. It seems useful to specify it even if very often DEBIRI is used to indicate a TACE that uses any type of microspheres.

Thank you for pointing out that DEBIRI and DEBDOX are trademarks of Biocompatibles UK Ltd. and should only be used with DC Beads. Fixed in manuscript.

2.Another point to be better defined is the infusion rate. In materials and metheod what does it mean "slowly"? Pls report how many ml of solution are infused for each minute and what dilution the vial of Embozene has.

We added the infusion rate (approx. 1mn / min) and a detailed description of the preparation of the mixture of microspheres and contrast agent.

3.There are unclear terms: line 109-110 eve? surgey?

Fixed in manuscript.

4. 137: most? pls specify the number,

We have specified the number of days until the patient's discharge.

5. 247 forget lipiodol,pls specify differences c-TACE Deb-TACE?

We showed a difference between cTACE and DEB-TACE. Fixed in manuscript

6. References 15  tumordestruction pls detache .

Fixed in manuscript

7.The observation time at 1-7 days is too long between 1 and 7 because we see the increase in transaminases after 2-3 days, pls add data from 1 to 5 days.

I agree with your point. The PES data taken between days 1-5. On the 7th day after the procedure, we conducted a routine interview with all patients.

Reviewer 2 Report

jcm-1046134

Comments to Author
Major strengths of the manuscript: 
•       Sets out to describe the side-effect/complication risk related to site of DEBIRI TACE in the treatment of mCRC.
•       The authors describe for the first time that there is no significant relationship between level of DEBIRI TACE and the rate of observed complications. They did however find a significant relationship between level of treatment and severity of PES.

  • Generally well written.

Major weaknesses of the manuscript:
•     Retrospective observational study of only 49 patients.

  • The study does not address efficacy (outcomes) of DEBIRI TACE performed from different branch levels.

Implications for patient care:
•       The level at which DEBIRI TACE for mCRC is delivered is of clinical interest for interventional oncologists and clinicians taking care of these patients. The study attempts to explain the effect of embolization level on complications and PES.

Specific comments:
Abstract:

  • Generally well written.
  • Fails to report which specific level of embolization led to higher rates of PES.
  • Page 1, line 38: Should the word “embolisate” be changed to “embolizate”?

Introduction:

  • Well written and well cited.
  • No changes needed.

Materials and Methods:
•     Well written.

  • Page 3, Lines 98-99: Could the authors be more specific as to what “high risk of gastrointestinal bleeding” means? Are these patients with varices? Portal hypertension?
  • Page 4, lines 156-160: Is there a reference for the PES adopted scale?
  • Statistical analysis appears to be appropriate.

Results:
•     Well written.

  • Tables are appropriately positioned within the body of text.
  • Page 8, line 238: Should the word “embolisate” be changed to “embolizate”?

Discussion:
•     Well written and cited.

  • Page 9, line 256: What is meant by “cardiac episodes”?
  • Page 9, line 292: There is an unnecessary space before the period ( .) at the end of the sentence.

Figures and References:
•       References are well cited and representative of current literature. Please review the instructions for authors regarding the listing and format of references.

  • Table 1: No changes necessary.
  • Table 2: No changes necessary.
  • Table 3: May be unnecessary and could be excluded as it is reported well in the Results section.
  • Table 4: No changes necessary.
  • Table 5: No changes necessary.
  • Table 6: No changes necessary.
  • Figure 1: Would suggest cropping the image better.
  • Figure 2: Would suggest cropping the image better.
  • Figure 3: The Kaplan-Meier analysis may be unnecessary.

Author Response

Thank you for the critiques and suggestions.

1. Abstract:Fails to report which specific level of embolization led to higher rates of PES.

We clarified that the severity of PES increased from sub-segmental through segmental to lobar administration.

Page 1, line 38: Should the word “embolisate” be changed to “embolizate”?

Fixed in manuscript.

2. Materials and Methods:

Page 3, Lines 98-99: Could the authors be more specific as to what “high risk of gastrointestinal bleeding” means? Are these patients with varices? Portal hypertension?

We explained that there is a high risk of gastrointestinal bleeding from esophageal varices in the case of portal hypertension.

Page 4, lines 156-160: Is there a reference for the PES adopted scale?

We added reference in the text.

3. Results:

Page 8, line 238: Should the word “embolisate” be changed to “embolizate”?

Fixed in manuscript.

4. Discussion:

Page 9, line 256: What is meant by “cardiac episodes”?

We have listed cardiac episodes: myocardial infarction, arrhythmia.

5.Figures and References:

Table 3: May be unnecessary and could be excluded as it is reported well in the Results section.

We deleted tables 3

Figure 3: The Kaplan-Meier analysis may be unnecessary.

We deleted Kaplan-Meier analysis.

References have been reformated and updated.